



# Quality assessment of aerosol lidars at 1064 nm in the framework of the MEMO campaign

Longlong Wang[1], Zhenping Yin[1,*], Zhichao Bu[2], Anzhou Wang[1], Song Mao[1], Yang Yi[1], Detlef Müller[1], Yubao Chen[2], and Xuan Wang[1,3,*]

[1]School of Remote Sensing and Information Engineering, Wuhan University, 129 Luoyu Road, Wuhan 430079, China
[2]Meteorological Observation Center, China Meteorological Administration, 46 South Zhongguan Road, Beijing, 100081, China
[3]Wuhan Institute of Quantum Technology, Wuhan 430206, China

**Correspondence:** Zhenping Yin (zp.yin@whu.edu.cn), Xuan Wang (xuan.wang@whu.edu.cn)

**Abstract.** Aerosol lidar network can play an important role on revealing structural characteristics of atmospheric boundary layer, urban heat island effect, spatial distribution of aerosols, especially monitoring atmospheric pollution in a megacity. To fulfill the need of monitoring and numerical forecast of atmospheric pollution, an aerosol lidar network is proposed by China Meteorological Administration, which serves as an important part of "MegaCity Experiment on Integrated Meteoro-

logical Observation in China" (MEMO). To ensure high standard of data quality and traceability of measurement error, an inter-comparison campaign, dedicated for quality assessment of lidar systems from different institutes and manufacturers, was designed and performed at Beijing Southern Suburb Observatory in September 2021. Six Mie-Rayleigh lidar systems at 1064 nm were involved in this campaign. The strategies for lidar self-evaluations and inter-comparisons were predefined. A lidar system at 1064 nm, which was developed by Atmospheric Remote Sensing group at Wuhan University, was selected as

the reference lidar system after passing all self-evaluations quality checks in a strict way. The reference lidar system serves as the corner stone for evaluating the performance of other lidar systems. After using the self-test of Rayleigh fit and signal-to-noise evaluation for each individual lidar system to fast check the data quality, the range-corrected signal and backscatter coefficient obtained from all the lidar systems were inter-compared with a reference lidar system. In the end, the lidar systems were quality assured, of which the standard deviation of range-corrected signal can be controlled within 5 % at 500-2000 m

while 10 % at 2000-5000 m. For the derived aerosol backscatter coefficients, standard deviations can be controlled within 10 % at 500-2000 m and 2000-5000 m. The quality assurance strategy lays down a solid basis for atmospheric lidar at near-infrared wavelength and will be applied in Chinese lidar network development.

## 1   Introduction

Atmospheric lidar plays a crucial role in observing the Earth's atmosphere. It enhances our understanding of the roles that

clouds and aerosols play in our climate system by providing high temporal and vertical spatial resolution profiles of aerosol backscatter coefficient, extinction coefficient and other intensive optical properties (Sugimoto and Lee, 2006; Müller and et.al., 2007; Mona and et.al., 2012). In addition, it has been utilized to observe the vertical distribution of water content (e.g.





Whiteman, 2003; Liu et al., 2022), temperature (e.g. Hauchecorne and et.al., 1992; Weng et al., 2018), cloud layers and cloud phase (e.g. Haarig et al., 2016; Lolli et al., 2018; He et al., 2022), and in particular aerosol distributions and characteristics
(such as smoke plumes, properties and transport of mineral dust aerosols, marine aerosols and other pollutants (e.g. Papayannis et al., 2009; Engelmann et al., 2021; Groß et al., 2011; Qin et al., 2016; Mamouri and Ansmann, 2017; Wang et al., 2019a; Yin et al., 2019; Wang et al., 2019b; Yin et al., 2021; Wang et al., 2022) in the atmosphere.

For the investigation of long-range aerosol transport mechanisms, it is necessary to extend the scale of investigation region, which is achievable by establishing a large ground-based lidar networks such as the EARLINET (European Aerosol Lidar
NETwork, D'Amico et al., 2015) in Europe, which is a part of the Aerosol Cloud and Trace Gases Research Infrastructure (ACTRIS) and also PollyNET (POrtabLe Lidar sYstem NETwork, operated as a part of EARLINET, Baars et al., 2016), AD-NET (Asian Dust and aerosol lidar observation network, Sugimoto and Lee, 2006) in Asia, as well as MPLNET (Micro-Pulse Lidar NETwork; Welton and et.al., 2001 and Lolli et al., 2019) around the world. To obtain a quantitative, unbiased, quality assured and statistical-wise dataset of lidar observations, the lidar instruments must be consistent in their performance
after being deployed at long-range distributed multiple stations. Therefore, to ensure the accuracy and consistency of the dataset is a crucial issue for the reliability of a lidar network. However, lidar systems are rather complex, containing several subsystems, which are not easily standardized, and their performance is critically dependent on a number of adjustments. Lidar calibration is one of the main processes used to ensure instrument accuracy. In one method of calibration, measurements are compared between an un-calibrated lidar and a reference instrument, which is used to check the accuracy of lidar products.
There have been various methods used for calibrating a lidar. MPLNET was calibrated by normalizing their signal to the molecular profile but requires knowledge of the aerosol optical depth of the atmosphere to correct the transmission loss of laser power (Welton and et.al., 2001; Lolli et al., 2019). Since the MPLNET lidar sites are co-located with AERosol RObotic NETwork (AERONET) (Holben and et.al., 1998) sites, the aerosol optical depth can be derived directly from the AERONET column optical depths measured by a sun-photometer. A comprehensive method for self-checking lidar hardware was proposed
by EARLINET Freudenthaler et al. (2018). While to achieve comparable performance at many stations, EARLINET used to perform direct inter-comparisons at the system level (Grabbe et al., 1996; McDermid et al., 1990; Ferrare et al., 1995; Sherlock et al., 1999; Freudenthaler et al., 2010). In addition, It has established the Lidar Calibration Centre (LiCal) using the reference lidar system to calibrate and assess other lidars and ceilometers (Matthais et al., 2004; Böckmann et al., 2004; Sicard et al., 2009; Pappalardo et al., 2014; D'Amico et al., 2015; Wandinger and et.al., 2016; Papagiannopoulos et al., 2016; Proestakis
et al., 2019; Campbell et al., 2002).

Since European lidar technology was independently developed at different stations from different countries, the devices and algorithms used are not the same (D'Amico et al., 2015). The generally accepted way to check the quality and reliability of lidar performance is to place many lidar systems for co-located measurement and data comparison simultaneously. Once good consistency from multiple lidar data obtained, all of these lidars are considered accurate (Matthais et al., 2004; Böckmann et al., 2004; Sicard et al., 2009). In 2000, there were nineteen lidar sites from eleven countries in Europe to build the EARLINET. In
2009, EARLINET organized a lidar inter-comparison campaign, where eleven systems from nine sites were jointly compared in Leipzig, Germany (Freudenthaler et al., 2010). After that, the other lidars were calibrated using the system that had been





compared, and the whole inter-comparison process was completed in 2013 (Papagiannopoulos et al., 2016; Proestakis et al., 2019). Finally, the deviation of lidar returned signal is less than 2 %, the deviation of boundary layer aerosol backscattering

signal is less than 10 %, and the average deviation is less than 5 % (Wandinger and et.al., 2016). On the basis of mutual calibration, the EARLINET has established three calibration centers in Italy, Romania and Germany respectively for lidar calibration (D'Amico et al., 2015; Pappalardo et al., 2014).

The American MPLNET Observatory has been online since the 1990s and now has eighty-two sites around the world. Since 2000, MPLNET has used a standardized lidar system for networking calibrations, and all of them use unified automated data

analysis algorithms. The overall hardware of MPLNET is produced by the manufacturer, and preliminary hardware verification is completed (Campbell et al., 2002). It was then transported to the GSFC center for testing and inter-comparison with a standard lidar (Córdoba-Jabonero et al., 2021).

The Mie-Rayleigh lidar at 1064 nm is commonly used for aerosol in particular smoke and volcanic ash as well as cirrus (Haarig et al., 2018; Vaughan et al., 2019; Pauly et al., 2019; Li et al., 2020; Yuan et al., 2022; Haarig et al., 2022) due to

its higher atmospheric transmission than that at 532 nm (Salvoni et al., 2021; Wu et al., 2020; Liang et al., 2019; Xian et al., 2020; Xu et al., 2022). Accurate measurement of aerosol/cloud backscatter coefficient at 1064 nm are critical to improve our understanding of various physical properties of the atmosphere, specifically how clouds and aerosols radiatively impact our Earth in the infrared (Pauly et al., 2019). Due to the weaker molecular signal-to-noise ratio (SNR) at 1064 nm compared to 532 nm for these instruments, calibration for 1064 nm attenuated total backscatter (ATB) calibration are based on the 532 nm

ATB calibration (Vaughan et al., 2019). The 1064 nm signal is calibrated utilizing the 532 nm calibrated signals within cirrus clouds. For any individual profile, the CALIPSO at 1064 nm calibration coefficient is simply the product of the interpolated instantaneous value of the scale factor time history and the corresponding calibration coefficient at 532 nm. However, the backscatter lidars at 1064 nm to fill the existing observational gaps within the existing lidar networks at the global scale are in continuous growth due to the advantages of low-cost, unattended and continuous operation, while the study on quality control

and assessment of their hardware and data is still very limited reported (**?**).

In order to quantitatively assessment of the direct impact of aerosol concentration on air quality and exclude the interference of meteorological factors, it is important to deeply understand the frequent outbreaks of long-term air pollution in large regions (such as Beijing-Tianjin-Hebei region and its surrounding areas). At present, China is starting to build a comprehensive and stereoscopic observation network (Lv et al., 2020; Huang et al., 2019; Chen et al., 2019), which leads to the isolated and one-

sided measurement of each stereoscopic observation station, therefore qualitative analysis of the spatio-temporal association between the stations with the non-biased, comprehensive, and statistical dataset is urgently needed. With the gradual shift from qualitative measurement to the qualitative application of atmospheric lidar, the requirements for its instrument function and data quality are increasingly high, and direct mutual comparison must be made at the system level, China has made great efforts to mitigate its long-standing environmental problem in recent years. There are many aerosol lidar observation stations

in China, which are mainly polarization Mie-Rayleigh lidars at 532 nm. In 2017, Chen et al. (2019) carried out self-calibration and inter-comparison experiments of multiple aerosol lidar for the first time in Beijing. Subsequently, an inter-comparison experiment involving twelve lidar systems was carried out using a reference lidar system (REAL-VIS at 532 nm) in 2019.



At present, China lidar calibration methods mainly focus on the influence of system performance on aerosol backscattering coefficient retrieval at 532 nm, thus the calibration and inter-comparison of 1064 nm channel are not only missing in China, but also rarely reported in the other region.

Based on the lidar inter-comparison observation campaign on September 2021 in the south of Beijing observatory, this paper introduces the lidar quality assessment strategy based on experience of EARLINET on self-calibration and inter-comparsion methods for systematic improvement of lidar hardware, and evaluates the reliability of the 1064 nm channel of many sets of lidar systems, analyzes the deviation of the Mie-Rayleigh signal and its influence on the backscatter coefficient. It aims to provide a relatively comprehensive quality assessment and control scheme for a single wavelength lidar system at 1064 nm, optimize the data consistency comparison and verification scheme among multiple lidar systems, and quantitatively evaluate the errors of data products, so as to lay a foundation for the subsequent establishment of a long-term and stable megacity aerosol lidar observation network. The campaign methodology and the results are discussed in the following. In section 2, an overview of the campaign with the description of involved lidar systems and the strategy applied is given. In section 3, the self-test and inter-comparison results are presented. In section 4, a discussion is provided based on the results on lidar signal and aerosol products. Finally, section 5 gives the conclusions and an outlook for future work.

## 2 Methodology

### 2.1 Intercomparison campaign overview

This inter-comparison campaign was carried out at the Nanjiao observatory at the southern outskirts of Beijing (39.95° N, 116.39° E; 39 m a.s.l.) on 27 September, 2021, which was organized by China Meteorological Administration, as one part work in the framework of the MEMO campaign. During the campaign, the weather is calm with wind speed less than 3 m/s.Cirrus is present and covers the height range from 7 to 12 km at most of the time. For the campaign, six co-located lidar systems with infrared channels were involved. These instruments were manufactured by different specialized companies and feature with different configurations, including different transmitting and receiving modules. The ID numbers were made up for these lidar systems at 1064 nm in order to easier identify and their hardware parameters provided by their manufactures are summarized in the Table 1. As it is very difficult to construct a mean signal as an absolute reference for the inter-comparison, the lidar at 1064 nm developed by Wuhan University was employed as a reference system, because each of its components was well-characterized and system was well-calibrated using the EARLINET quality assurance standards (Freudenthaler et al., 2018) as well as inter-compared respect to the standardised lidar at 532 nm (Chen et al., 2019). It has ability to observe the atmosphere at the range from 0.2 km up to more than 10 km in the nighttime conditions at 1064 nm with a repetition rate of 2500 hz and 0.1 mJ laser emitter and a diameter of 200 mm f/2.5 Cassegrain telescope receiver. The 1064 nm light is extracted by an interference filter at the center wavelength 1064.2 nm with 1 nm bandwidth provided by Alluxa Inc. (https://www.alluxa.com/). In order to eliminate the background noise from the detector, the backscattering signal is collected by a single photon avalanche photodiode (SPAD) detector (SPCM-AQRH-13, Excelitas Canada Inc., https://www.excelitas.com/product/spcm-aqrh) with photon counting mode and its output signal is amplified and digitized by the configurable lidar acquisition system (CLASS,



advanced lidar applications s.r.l, https://alasystems.it/) with spatial resolution of 15 m. In this campaign, all lidar systems employed a SPAD detector with photon counting mode except for the No.L05 lidar system, of which the analog detection mode of avalanche photodiode detector (APD) was applied. The inter-comparison observation includes two parts. The first part is self-validation or calibration according to EARLINET quality assurance tool (Freudenthaler et al., 2018), which includes Rayleigh-

fit, detectable range check as well as telecover test. The second part is inter-comparison with the measurements taken from reference lidar respectively, as can be seen in Figure 1. With the CMA's goal of promoting the use of lidar instruments and their data among the Chinese lidar network, the inter-comparison at the hardware level was made, in terms of range corrected lidar return signals inter-compared directly, and also the inter-comparison of aerosol backscatter coefficient at 1064 nm retrieved by each lidar system was performed in this study.

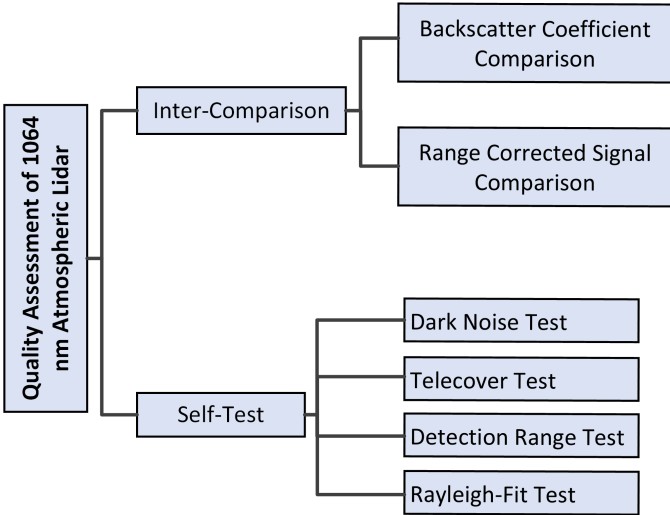

**Figure 1.** Diagram of quality assessment strategy for 1064 nm atmospheric lidar.

**Table 1.** The ID numbers of lidar systems at 1064 nm and their hardware parameters provided by the manufactures.

| ID number | receiver diameter [mm] | energy [mJ] | frequency [Hz] | Detection mode |
|---|---|---|---|---|
| Ref. | 200 | 0.1 | 2500 | Photon Counting |
| L01 | 100 | 0.1 | 1000 | Photon Counting |
| L02 | 40 | >0.1 | 2500 | Photon Counting |
| L03 | 160 | >0.1 | 1000 | Photon Counting |
| L04 | 250 | 1 | 1000 | Photon Counting |
| L05 | 200 | 30 | 20 | Analog |



## 2.2 Intercomparison strategy

In the entire measurements from all the lidar systems, the raw data were sampled with a time resolution of 1 minute and range resolution of 15 m so that makes it easier for inter-comparison. The measurement data was re-formatted and processed by Atmospheric Lidar Evaluation program (ALiE, https://gitee.com/mualidar/cma-lidar-comparison) after some basic configurations. The assessment report and figures can be generated automatically to assist the lidar performance evaluations. The program structure can be found in Figure 2

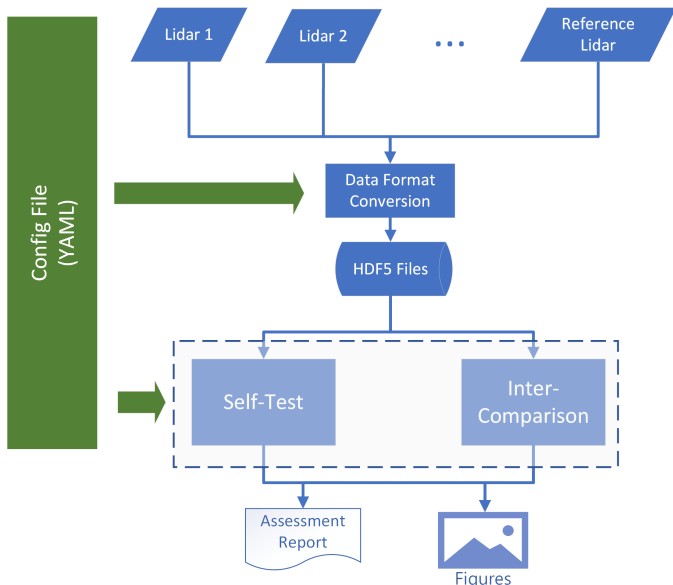

**Figure 2.** Software structure of Atmospheric Lidar Evaluation program for 1064 nm atmospheric lidar inter-comparison, in which "YAML" standards for human-readable data-serialization language and was widely used for cross-platform software configurations. "HDF5" stands for Hierarchical Data Format, which was commonly used for atmospheric data storage.

All the measurement data was pre-processed before being used in self-test and inter-comparisons. Within pre-processing, systematic effects were corrected, for instance, dead time and pre-trigger corrections. each single system has made the self-calibration of Rayleigh fitting with the calculated molecular attenuated backscatter coefficient using the standard atmospheric model according to Freudenthaler et al. (2018) in order to quickly evaluate detectable ability each system, and also its signal-to-noise ratio (SNR) was assessed to estimate the detectable range using the existed method (Morille et al., 2007). Before the inter-comparison of each lidar signals was made, all the range-corrected signals (RCS) were normalized based on the signal obtained from the reference lidar system in order to avoid biases due to the difference in each system efficiencies. Normalized range corrected signal $z_i^2 P_{\mathrm{norm}}(\lambda_r, z_i)$ was obtained by fitting with the respective received powers from altitude





$z_i\,P_{\mathrm{norm}}(\lambda_0, z_i)$ at reference range $z_{min}$ to $z_{max}$ after the subtraction of solar background noise as

$$z_i^2 P_{\mathrm{norm}}(\lambda_r, z_i) = z_i^2 P_i(z_i) \frac{\sum\limits_{z_{min}}^{z_{max}} z^2 P_{\mathrm{ref}}(z)}{\sum\limits_{z_{min}}^{z_{max}} z^2 P_i(\lambda_r, z)}. \tag{1}$$

It should be compared with simultaneous observation, and continuously collecting the original data for at least 180 minutes, and selecting a period of no less than 30 minutes where the aerosol vertical distributions are stable. The rang-square-correction signal is obtained. The original signal profile was obtained by using the original collection data, and the cumulative average, background subtraction, and rang-square-correction were carried out on the original signal profile. After obtaining the normalized signal of rang-square-correction, the aerosol backscattering coefficient was calculated according to the algorithm from Fernald (1984). The selected reference height interval is 6000-6500 m, and the fixed lidar ratio of 50 sr is adopted. Using the rang-square-correction or aerosol backscattering coefficient from No.Ref. lidar as the reference signal, the relative deviation of the profile from the other lidars are calculated according to equation 2 :

$$\delta = \frac{S_{\mathrm{t}} - S_{\mathrm{ref}}}{S_{\mathrm{ref}}} \cdot 100\% \tag{2}$$

which $\delta$ is the relative deviation between two lidar systems; $S_{\mathrm{ref}}$ is the reference lidar signal, $S_{\mathrm{t}}$ is the lidar signal to be compared. Then, in order to assess the lidar performances within the dense aerosol loading and relatively aerosol-free region, their averaged relative deviation was calculated by integrating the profile signals within 0.5-2 km (dense aerosol loading) and 2-5 km (aerosol free) respectively:

$$\bar{\delta} = \frac{\sum_{i=1}^{n} |\delta|}{n} \cdot 100\% \tag{3}$$

which $\bar{\delta}$ is the averaged relative deviation within 0.5-2 km and 2-5 km, and $n$ is the sampling points in the selected height intervals.

## 3 Results

### 3.1 Self-test results

Each lidar signal at 1064 nm is performed using the Rayleigh fit test (Figure 3). In order to avoid the possible effects caused by saturation of detectors, in this study, we decided to fit the molecular attenuated backscatter coefficient to the RCS of each lidar system. The normalization range was chosen to be from 6000 m to 7000 m, which is in an aerosol-free region and still with a good signal-to-noise ratio except No. L05 lidar adopted the normalization range between 12000 m to 13000 m due to its signal issue at such range. The mean relative deviation of all Rayleigh fits within the normalization range were found to be less than 5 % (Table 2), which indicated the good agreements between all the lidar signals and the atmospheric molecular attenuated backscattering coefficient in the free atmosphere. As Figure 3 (a-e) shows, all the lidar signals were found that can present the near real atmospheric molecular backscattering in an aerosol-free region, which is about from 3000 m to 7000 m.





However, the signal of No.L05 lidar (Figure 3, f) was found to float up with the increased range above 3000 m, therefore the real atmospheric molecular backscattering can be presented with this system. We assumed such problem was caused by the electronic noise from the analog detector.

**Figure 3.** Rayleigh fit (orange) to the normalized rang-square-correction signals for the range correct signals (RCS) of each lidar systems at 1064 nm (grey) during 18:00 to 18:30 China Standard Time (CST) on 27 September 2021. Horizontal light blue dash lines indicated the selected aerosol-free region. (a) RCS from the reference lidar system, (b) RCS from the NO.L01 lidar system, (c) RCS from the NO.L02 lidar system, (d) RCS from the NO.L03 lidar system, (e) RCS from the NO.L04 lidar system, (f) RCS from the NO.L05 lidar system.





**Table 2.** The mean relative deviation of Rayleigh fit for each lidar system, MRD is indicated to mean relative deviation within each selected normalization range (1000 m range interval).

| Lidar ID | Ref. | L01 | L02 | L03 | L04 | L05 |
|---|---|---|---|---|---|---|
| **MRD [%]** | 2.4 | 4.3 | 3.7 | 3.6 | 1.9 | 3.0 |

The SNR was analyzed as well in order to check the detectable range of each lidar system. In this test, each single profile was averaged with 30 minutes intervals. The background noise was calculated by the last 50 range bins of signal, and the SNR was calculated according to the method described by (Morille et al., 2007). We defined the lidar signals are valid when the value of SNR is larger than 3. The maximum detectable range of the lidar systems was found to be over 7000 m (Figure 4, a-e) except for lidar No.L05 which was effected by the problem with noise over 5 km (Figure 4, f). It also can confirm the assumption that

lidar No.L05 has a problem with the noise.

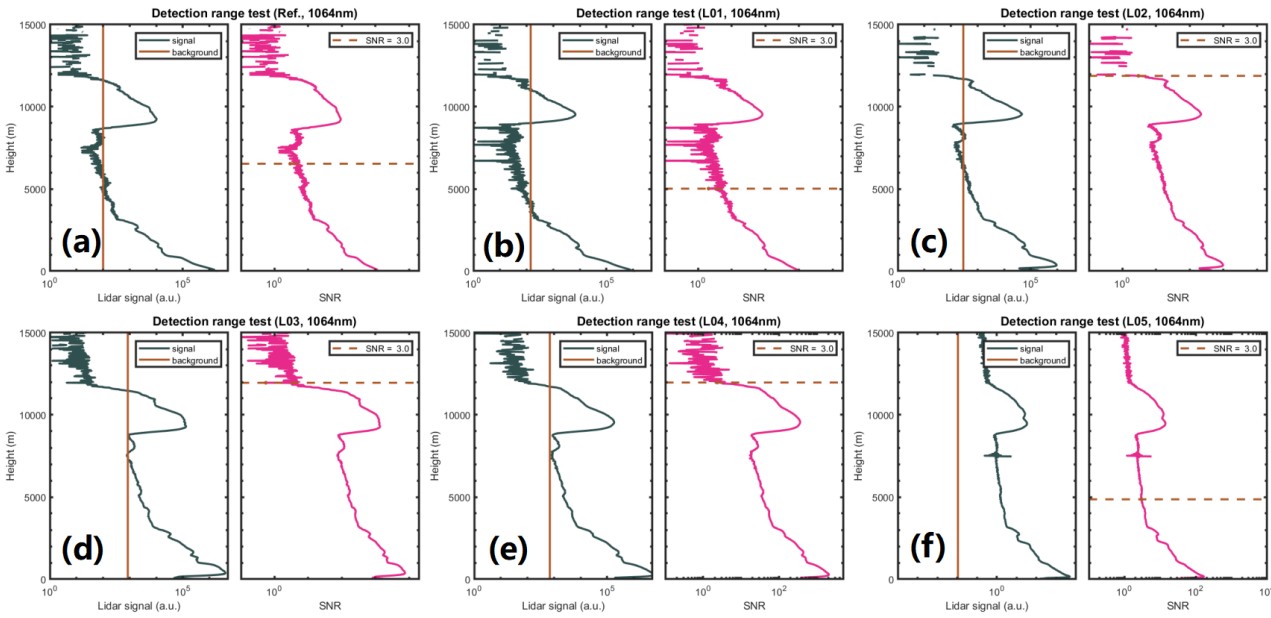

**Figure 4.** The detectable range test results of raw signals at 1064 nm from each lidar system during the same period as Rayleigh fit (Figure 3). The black curves represent the signals with its background. The orange vertical lines indicate the value of the background from each lidar systems. The pink curves represent the signals with its signal-to-noise ratio (SNR=3) region. the orange horizontal dash lines indicate where the SNR is equal to 3. (a) Raw signal from the reference lidar system, (b) Raw signal from the NO.L01 lidar system, (c) Raw signal from the NO.L02 lidar system, (d) Raw signal from the NO.L03 lidar system, (e) Raw signal from the NO.L04 lidar system, (f) Raw signal from the NO.L05 lidar system.





## 3.2 Inter-comparison results

The data was collected continuously over at least three hours so that it is able to select a 30 minutes period with calm weather and stable aerosol distribution conditions. Figure 5 showed each lidar signal performance on aerosol loading distributions on 27 September 2021 between 15:00 and 23:00 CST. As a quick look, a good consistences were found between all the lidar measurements, such as the almost same height of cirrus clouds bottoms (above 8000 m) and tops (around 11000-12000 m) as well as the aerosol distributed under about 3000 m. The results, therefore, indicated that the output of each lidar system are comparable qualitatively, which also means they can be used to observe the vertical distributions of aerosol and cloud without knowing the determining the amounts.

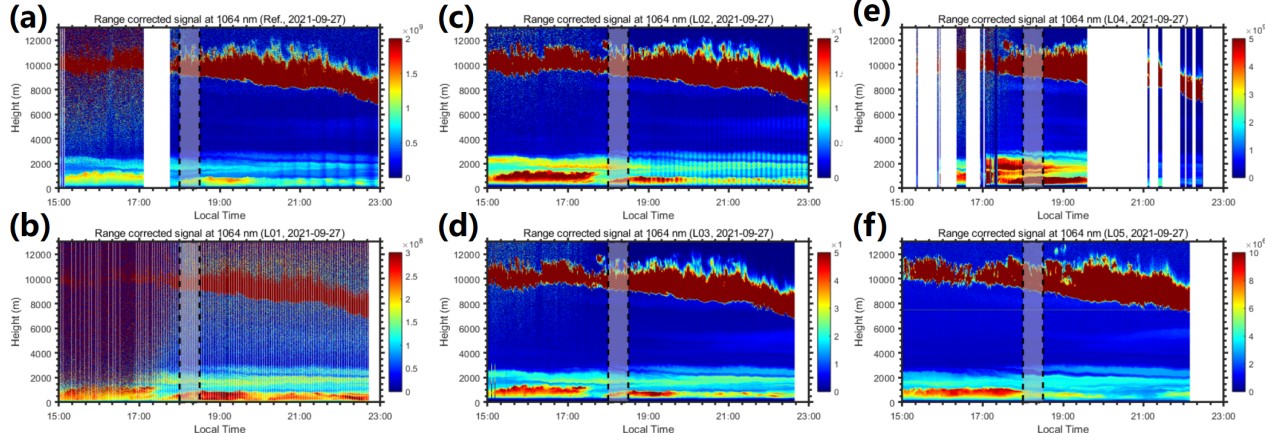

**Figure 5.** Quick looks of aerosol loading observation from different lidar measurements on 27 September 2021 between 15:00 and 23:00 CST. The grey rectangle shadows indicate the selected periods for single profile inter-comparison between 18:30 and 19:00 CST. The RCS at 1064 nm were taken from (a) reference lidar system, (b) L01 lidar system, (c) L02 lidar system, (d) L03 lidar system, (e) L04 lidar system (the multiple interruptions were caused by the operation issue during the measurement) and (f) L05 lidar system. All data was sampled in 1 min temporal resolution and to 15 m range resolution.

In order to quantitative analysis the inter-comparison results, the normalized RCSs were inter-compared and the accuracy of each lidar system were presented. Due to the different overlap properties in the near range and multiple scattering properties in the clouds, the inter-comparison of RCSs were mainly investigated in the region of 500-2000 m (the main aerosol layer) and 2000-5000 m (the clear atmosphere expected). In this investigation, the single profile by 30 minutes averaging during 18:30 and 19:00 CST was selected. The window range from 1500 m to 2000 m was adopted to normalize all the RCSs where the aerosol loading is relatively stable. As Figure 6 (a) shows, the good consistences were found between all the RCSs for the detection of cirrus cloud base about 8000 m and aerosol loading below 3000 m. As the results are shown in the Rayleigh test, all the lidar systems performed the reliable detection ability on the relative atmospheric clean region between about 3000 m and 8000 m except for lidar No.L05. Because of the huge difference under the in-completed overlap region between different





lidar systems, the large relative deviations were shown under 500 m (Figure 6, b)), thus it is not able to compare. While the relative deviations between lidar No.L01-L04 and the reference lidar were under 50 % from the range 500 m to 12000 m, the

relative deviations were found less than 20 % in the range from 500 m up to 12000 m and less than 5 % in the range from 500 m up to 5000 m between lidar No.L02-L04 and the reference lidar. The overestimation of No.L01 was found to be about 20-40 % above 3000 m and the underestimation of No.L01 was found to be about 40 % in the cirrus, which is probably due to its poor SNR in the higher range or multiple scattering effects by the cloud layer. The lidar No.L05 had a larger overestimation (over 50 %) with the reference lidar as well as the others in the aerosol-free region, but underestimate the aerosol loading and cirrus

about 20-50 %. The mean relative deviations within 500-2000 m and 2000-5000 m were found to be less than 5 % and 10 % respectively (Figure 6, c) except for lidar No.L05.

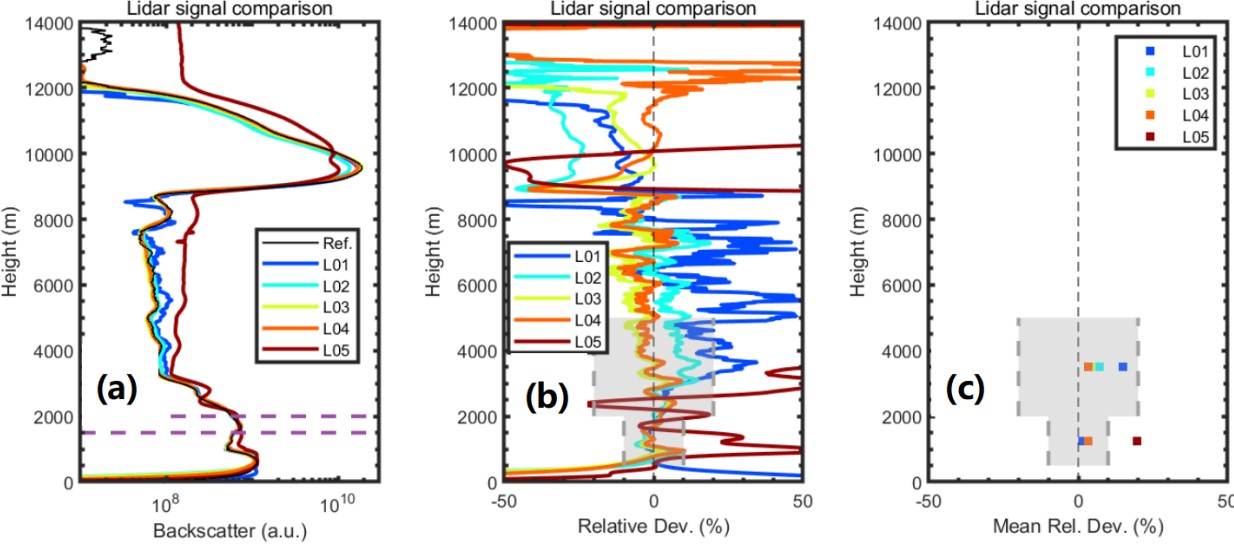

**Figure 6.** The inter-comparison results for RCSs from all the lidar systems in the periods of 18:30 and 19:00 CST on 27 September 2021. The horizontal dash lines indicate the adopted normalization window for each RCSs. The vertical dash lines in the middle indicate zero relative deviation. The rectangle shadows indicate 10 % of relative deviation within 500-2000 m and 20 % of relative deviation within 2000-5000 m. (a) normalized RCS profiles, (b) the profiles of relative deviations of lidar No.L01-L05 with the No.Ref., (c) the mean relative deviations of lidar No.L01-L05 with No.Ref..

In order to find the results of uncertainties propagated from the RCSs to the aerosol optical products, the Fernald algorithm for all the RCSs was used with the same assumed fixed lidar ratio of 50 sr, and also the same reference height of aerosol-free region (6-6.5 km) for aerosol backscatter coefficient retrieval was selected. The RCSs were taken the same as Figure 6, but the

range resolution of all RCSs was re-sampled to be 100 m. The aerosol backscatter coefficient profiles at 1064 nm were shown in Figure 7 (a) under 7000 m, and the relative deviation profiles of No.L01-L05 with No.Ref. were analyzed respectively as shown in Figure 7 (b). The relative deviations of aerosol backscatter coefficient at 1064 nm from lidar No.L01-L04 with the reference lidar were within 10 % from the range 500 m to 5000 m in general, while large relative deviations (about -50 %)





were found between lidar No.L05 and the reference lidar in the range from 500 m up to 5000 m. It also can be seen that both

maximum mean relative deviation in the height range of 500-2000 m and in the height range of 2000-5000 m were less than

10 % in Figure 7 (c). Compared with the relative deviations of RCS, there is no evident increase of relative deviations after the

aerosol backscatter coefficient retrieved by taking the same algorithm from the No.L01-L04 lidar systems, but larger relative

deviations were found between its RCS and aerosol backscatter coefficient from No.L05 lidar. Therefore, it may indicate the

relative deviations can be amplified if the lidar signal got severe distortion.

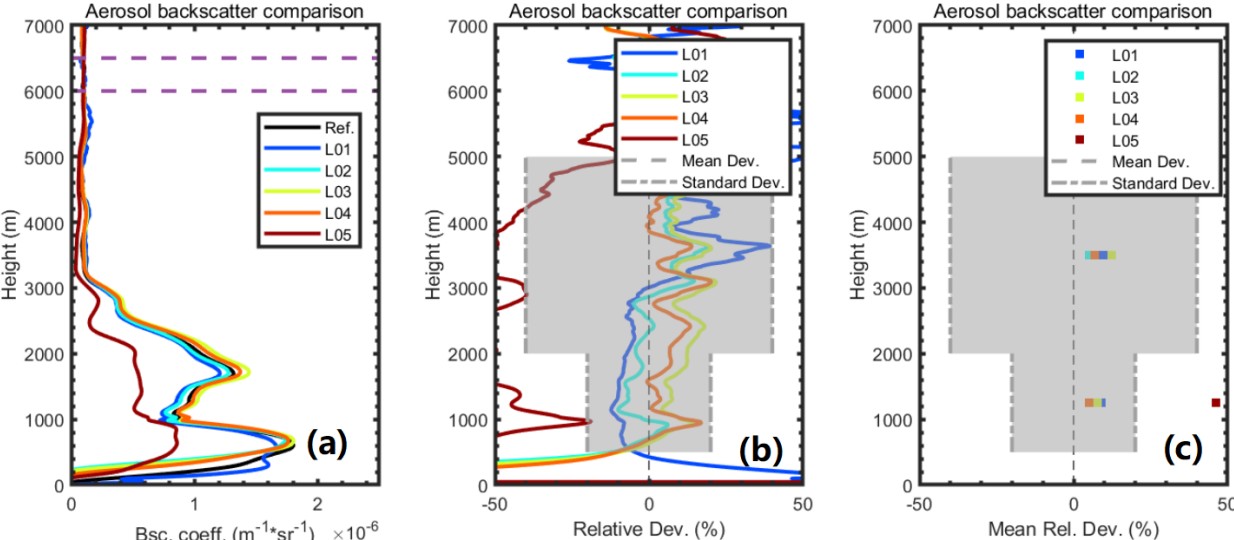

**Figure 7.** The inter-comparison results for aerosol backscatter coefficient at 1064 nm from all the lidar systems in the same periods with Figure 6. The horizontal dash lines indicate the referred aerosol-free window used for retrieval algorithm. The vertical dash lines in the middle indicate zero of relative deviation. The rectangle shadows indicate 10 % of relative deviation within 500-2000 m and 20 % of relative deviation within 2000-5000 m. (a) the aerosol backscatter coefficient profiles at 1064 nm, (b) the profiles of relative deviations of the aerosol backscatter coefficient profiles from lidar No.L01-L05 with its from No.Ref., (c) the mean relative deviations of the aerosol backscatter coefficient profiles from lidar No.L01-L05 with its from No.Ref..

## 4   Discussions

Using the EARLINET instrument inter-comparison strategy as a reference, the lidar inter-comparison observations were carried

out to provide a basis for megacity aerosol lidar networking. The self-test of the Rayleigh test and detectable range test are

able to evaluate the lidar system performances on long range detection, however, the quantitative accuracy of aerosol loading

and cloud bases still relies on the inter-comparison. The near-infrared channel of aerosol lidar usually suffers from the low

sensitivity or high electronic noise of the analog detectors so that it cause the severe distortion of atmospheric observation

particularly in the aerosol-free region (Sicard et al., 2009) as the example shown in Figure 3 (f), Figure 6 and Figure 7. In this

inter-comparison, five of six lidar system employed the photon counting mode using the SPADs. In such cases, we expected





the interference of electronic noise in the signals can be solved. In order to verify our assumption, the dark measurements
were performed for both photon counting SPAD mode and analog APD mode detectors (Figure 8). The reference lidar system

was selected as an example of SPAD performance to compare with the APD performance. The background noise from SPAD
detection showed random distribution around zero with the range (Figure 8, a), and the random noise is smaller than system
noise, which means the signals have fewer effects by the noise caused by the detectors. On the contrary, the background
noise from APD detection showed not only a decreasing structure with the increasing range above zero (Figure 8, b) but also
the system noise caused by strange structures with the range is twice bigger as random noise caused by the sampling time, in

addition to the sharp peak noise existed around 7500 m. Therefore the big distortion could not be ignored if directly to eliminate
the background noise using the tail of the raw signals (Freudenthaler et al., 2018). Despite the poor performance of NO.L01
over 4000 m, it is probably due to the misalignment between the telescope and laser directions, which requires further check
using the telecover test. From the performance on each lidar signals at 1064 nm, we also found there might be still space for
improvement of detection capabilities, while maintaining its miniature design, for instance, No.L02 lidar with 40 mm diameter

receiver and 0.25 W laser energy presented almost similar detection ability as No.L04 lidar with 250 mm diameter receiver and
1 W laser energy (See in Table 1). Thus, it seems the single channel aerosol lidar systems at 1064 nm are promising to be even
more compact and miniaturization.

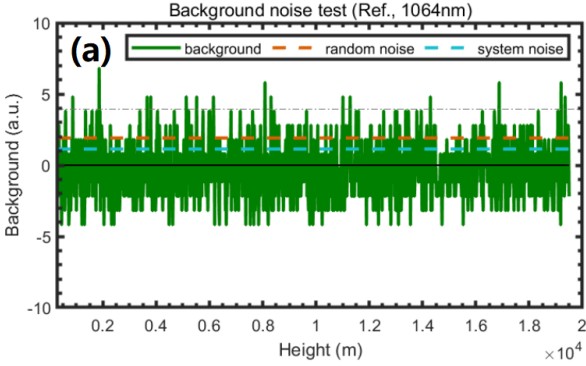
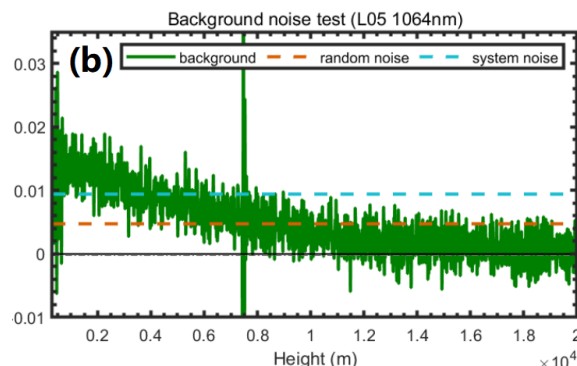

**Figure 8.** The background noise test results of the photon counting SPAD and analog APD detectors. The green lines indicate the background
noise directly collected by lidar system. The orange dash lines was the standard deviation of background noise, which was defined as random
noise. The sky blue dash lines was the maximum deviation of the mean background noise within a 500 m range widow to the mean background
noise of all data points, which was defined as system noise. The horizontal black lines indicate the value of zero. (a) the test result from lidar
No.Ref., (b) the test result from lidar No.L05.

## 5   Conclusions

In September 2017, the Mie-Rayleigh scattering channel at 1064 nm of six aerosol lidars were participated in MEMO lidar

inter-comparison campaign to be evaluated and calibrated at the Beijing Southern Suburb Observatory. A reference lidar at
1064 nm is used to construct a signal as an absolute reference for this inter-comparison, and also the unified algorithm was





adopted to process the raw lidar signals. Although the self-test, such as the Rayleigh fit test and SNR evaluation, can be used as fast signal check in the far range, the direct lidar inter-comparisons at 1064 nm is still very necessary and an efficient way to quantitatively assess the lidar performances on the dense aerosol distributed regions. In this campaign, a good agreements

of RCSs and backscatter coefficients at 1064 nm were obtained with the defined references using photon counting detection mode of the SPADs except those relative deviation of No.L05 lidar with analog detection mode of APD is still higher. The profiles of relative deviation of lidar signals are less than 5 % within 500-2000 m and 10 % within 2000-5000 m except that of No.L01 is higher which is probably due to the miss-alignment; The mean relative deviation of lidar signals within 500-2000 m and 2000-5000 m are a little lower than the profiles of relative deviation; The profiles of relative deviation of aerosol products

(backscatter coefficient) are slightly higher than those of lidar signals within 500-2000 m and 2000-5000 m using the unified algorithm, and the similar performances were found in the mean relative deviation of aerosol products within 500-2000 m and 2000-5000 m. However, the relative deviation of aerosol products also can be amplified from the lidar signals if it was large enough, such as the performance of No.L05. In general, the relative deviations of the above were found within the maximum boundary of permissibility proposed by EARLINET, thus it could gain enough confidence on reliability of the signals provided

by each lidar system in the channels at 1064 nm for future lidar network in China. In addition, we found the photon counting mode at 1064 nm could be a good solution of filling the observational gaps of backscatter coefficient at 1064 nm in a high accuracy and quantitative way, and also such lidar system has a possibility to be miniaturization. As it is the first report on lidar inter-comparison at 1064 nm in China, furthermore the inter-comparison of polarization channel at 532 nm, Raman channel at 386 nm and 607 nm for aerosol extinction coefficients, and the Mie-Rayleigh channel at 355 nm will be explored.

*Author contributions.*  All the authors made contributions to this research work and manuscript. In particular; Longlong Wang, Zhenping Yin and Xuan Wang designed the whole strategy of this work. Longlong Wang organized the observation campaign, analyzed the data, acquired the research funding, and wrote the manuscript draft. Zhenping Yin, Song Mao Yang Yi and Xuan Wang analyzed the data, participated in the scientific discussions, and reviewed and proofread the manuscript. Anzhou Wang and Zhichao Bu participated in the observations and data collections. Detlef Müller, Yubao Chen and Xuan Wang reviewed and proofread the manuscript. Xuan Wang and Yubao Chen acquired

the research funding and led the study All authors have read and agreed to the published version of the manuscript.

*Acknowledgements.*  This work was supported by the National Natural Science Foundation of China (grant Nos. 62105248 and 42205130). We are grateful to Darsun laser Technology Co., Ltd., China Aerospcae Science and Technology. Institute of Oceanographic instrumentation Shandong Academy of Science and so on., the manufacturers of the lidar instruments, which were provided by them for this study.

*Competing interests.*  The authors declare no conflict of interest. At the time of the research, the manufacturers of the lidar instruments,

which were provided data in the study, had no role in the design of the study; in the analyses, or interpretation of data; in the writing of the manuscript, or in the decision to publish the results.



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
