# Peer review of "Quality assessment of aerosol lidars at 1064 nm in the framework of the MEMO campaign"

_Atmospheric Measurement Techniques, 2023_

## Referee Comment (RC1)

**General comments**

This paper presents a comparison of lidar signals and the derived aerosol parameters between 6 different lidar systems using a inter-comparison strategy recommended by EARLINET. The objective is to evaluate the performance of different lidar systems and quantify their consistency. Such work is important to the scientific community and has to be done for lidar networking. However, such comparison is not easy to organize, because of the transportation of lidars, arrangement of the campaign, the strategy of data comparison and so on. The subject fits very well the scope of *<Atmospheric Measurement and Technique>*. It is a pity that (at least it seems) the campaign did not last long, since I did not see measurements on multiple days and with different aerosol loading, clean atmosphere and cloudless conditions. The presentations of results are well done and results are well analyzed.

As to academic writing, I suggest the author and co-authors spending more time on polishing the manuscript before getting it published. Be careful with the following 3 points:

- Remember to not use long sentence if you are not sure you can handle the grammar and logic in complicated sentences.
- Avoid colloquial expressions. In particular, be careful with the words such as 'maybe', 'probably', 'might' and so on. There are quite a few expressions like this throughout the manuscript. It is OK to have open questions or unresolved issues in a scientific paper, but you should state clearly what the causes/considerations are. Otherwise, it gives the readers an impressive that the results or conclusions are vague and informal and cannot be trusted.
- Make it shorter. I would expect this paper to be concise and compact. The author should focus on the main messages that he/she wants to address to the readers.

Moreover, I see it not necessary to have a separate section 'Discussion' with only the comparison of dark measurement. The discussion part is supposed to analyze and interpret the results of the study, but it is not the case here.

**Specific comments**

P1L13-15: "In the end, the lidar systems were quality assured, of which the standard deviation of range-corrected signal can be controlled within 5 % at 500-2000 m while 10 % at 2000-5000 m." → One example for consideration "In the end, the lidar systems passed the quality control/assurance, ensuring that the standard deviation of range-corrected signal could be controlled within 5% at 500-2000 m and 10% at 2000-5000 m"

P2L29: '  large ground-based lidar networks': remove 'a'

P2L45: '  to achieve', remove 'while'

P2L51-52: "Since European lidar technology was independently developed at different stations from different countries,"-> "Since lidar systems in EARLINET were developed independently at stations in different countries.." .

This sentence is not clear-- "the devices and algorithms used are not the same (D'Amico et al., 2015)". EARLINET has its requirements and criteria for all the affiliated lidar stations, as well

as data processing algorithm, the Single Calculus Chain. I am not sure what devices and algorithms you are referring to.

P3L80:  reference missing

P3L81: assessment of -> assess

P3L83: "At present, China is starting to build…" This sentence not clear, please revise it. Here is an example, for your reference-- "Currently, China is in the process of building a … network, which may lead to isolated and one-sided measurements from each observation station, therefore…spatial-temporal correlation…"

P3L86-87: "With the gradual shift from qualitative measurement to the qualitative application of atmospheric lidar, ... high, and direct mutual … level, China has" → As atmospheric lidars are shifting from qualitative to quantitative applications, …, increasingly high. Therefore, direct mutual…level. China has…

P4L95: in the other region → in other regions/countries.

P4L99:  This sentence is too long and lacks of clarity, please revise it. "Based on the lidar inter-comparison observation campaign on September 2021 in the south of Beijing observatory, this paper introduces the lidar quality assessment strategy based on experience of EARLINET on self-calibration and inter-comparison methods for systematic improvement of lidar hardware, and evaluates the reliability of the 1064 nm channel of many sets of lidar systems, analyzes the deviation of the Mie-Rayleigh signal and its influence on the backscatter coefficient."

P4L100: , …., single wavelength -> single-wavelength

P4L105:  on the lidar signal….

P4L110: , which was  -> . It was

P4L111:  Cirrus was… and covered

P4L112: in most of the time

P4L113: feature -> featured

P4L114: Maybe it is more accurate to say "emission and reception modules" instead of "transmitting and receiving modules ".

P4L121:  The 1064 nm light -> the backscattered 1064 nm light

P5L127: of which→ on which

P5L129: In this sentence "The first part is self-validation or calibration according…", what are the difference between (self-) validation and calibration? If these two terms are referring to the same thing, there is no need to use different terminology here, it could confuse the readers.  The authors should also check if there are confusions of terminology elsewhere in the manuscript, such as self-test, self-validation, calibration, verification… and so on.

P5L131: CMA not defined

P5L131-134:  "With the CMA's goal of promoting the use of lidar instruments and their data among the Chinese lidar network, the inter-comparison at the hardware level was made, in terms of range corrected lidar return signals inter-compared directly, and also the inter-comparison of aerosol backscatter coefficient at 1064 nm retrieved by each lidar system was

performed in this study." This sentence is not correct in grammar, please think of breaking long sentence into shorter ones for simplicity and clarity.

P6L137: "so that makes it …."-> This uniformity in data collection makes the inter-comparison easier.

P6L144: of each system.  Again, the sentence is too long and contains fragments.
P6L147:  due to the difference in each system efficiencies -> due to different lidar efficiencies/ transmissions

P7L150: Please specify what $\lambda r$, $\lambda 0$ and I are.
P7L151: please revise this sentence "It should be compared with simultaneous observation, and continuously collecting the original data for at least 180 minutes,
and selecting a period of no less than 30 minutes where the aerosol vertical distributions are stable."

P7L152: "The rang-square-correction signal" what is it? Range-corrected signal is commonly used in the lidar community.

P7L157: No. Ref lidar is confusing, better call it reference lidar for clarity.

P7L170: I do not see the link between signal saturation and fitting molecular signal to RCS. please explain.

P7L172: "…a good signal-to-noise ratio except No. L05 lidar adopted…". Break into 2 sentence "…a good signal-to-noise ratio. However, No. L05 lidar adopted….at such range"
P7L174: typo:  a space missing 'than' and '5%'
P8L177--L179: Could you give more comment about the data quality of lidar No.05?  Is it possible to correct this electronic noise? Could this mismatch with Rayleigh fit be caused by other reasons, for example, the divergence of laser beam?  Have you ever got better Rayleigh fit from this lidar?

P8L180:  Better to write "In this test, lidar profiles in every 30-minute time intervals were averaged."

P9 Figure 4:  To me, raw lidar signal means it is without any correction, such as deadtime, bin shift, etc. In Figure 9, I guess corrections have been applied.

P10L193: This is a bit confusing, please rephrase it "which also means they can be used to observe the vertical distributions of aerosol and cloud without knowing the determining the amounts".

P10L194: quantitative analysis→ quantitatively analyze
P10L194: it is not proper to call it 'accuracy', better to use 'difference'

P10L195: overlap properties may cause misunderstanding, you can simply say 'overlap ranges'

P10L197: "In this investigation, single…..was selected" → "Lidar observations between 18:30 and 19:00 CST from each lidar system were averaged for inter-comparison. "

P10L198: From the quicklooks, the readers can see the presence of aerosol layers in the range of 1500-2000, why did you normalize the lidar signal in the aerosol layer although you have mentioned in the previous sentence that the range 2000-5000 was expected to be clean? What if you normalize at higher altitude, should the results be better?

P11L202-L203: A few grammar mistakes in this sentence. Please rewrite it, here is one example "Due to the significant differences in the incomplete overlap region between different lidar systems, large relative deviations were observed within the 500 m range (Figure 6, b). As a result, a meaningful comparison cannot be made"

About P10 Figure 5:
- The ticks of Figure 5(c,d) are partly cut off, please replace with complete figures
- The structures within boundary layer looks similar in all quicklooks except for Figure 5(e), could you comment on that?
- Where do the repeating white stripes in Figure 5(b) come from? Is it a technical anomaly?
- In Figure 5(a) and 5(c), we see ripples in the transported aerosol layer below 3000 m from 19:00 to 23:00, can you comment on that?

P11, P12: Figure 6(c) and 7(c) do not carry so much information, therefore not necessary.

P13: Please write how you determine system noise and random noise and put references

The reviewer stops listing corrections of English writing from Page 11 and advices the author and coauthors to correct it by themselves.

Figure 8: (1) The scales of Y-axis in Figure 8(a) and Figure 8(b) are so different, making it difficult to conclude. (b) For Figure 8(b), did you take multiple dark measurements for L05 in order to check if such noises are stable or not?

P14L267: "we found … also such lidar system has a possibility to be miniaturization". It is not appropriate to draw such conclusion, because this paper does not talk about the miniaturization.

---

## Author Response (AR1)

*The authors would like to thank the editor for the help and reviewers for the thoughtful and helpful comments and suggestions, which have made a significant contribution to the improvement of the paper. We have considered all the comments and questions posed by the reviewer. They are listed one by one in this letter and implemented in the manuscript as text in blue color.*

To reviewer 1

*The authors would like to thank the reviewer for the thoughtful and helpful comments and suggestions, which have made a significant contribution to the improvement of the paper. We have considered all the comments and questions posed by the reviewer. They are listed one by one in this letter and implemented in the manuscript as text in blue color.*

**General comments**

This paper presents a comparison of lidar signals and the derived aerosol parameters between 6 different lidar systems using a inter-comparison strategy recommended by EARLINET. The objective is to evaluate the performance of different lidar systems and quantify their consistency. Such work is important to the scientific community and has to be done for lidar networking. However, such comparison is not easy to organize, because of the transportation of lidars, arrangement of the campaign, the strategy of data comparison and so on. The subject fits very well the scope of <Atmospheric Measurement and Technique>. It is a pity that (at least it seems) the campaign did not last long, since I did not see measurements on multiple days and with different aerosol loading, clean atmosphere and cloudless conditions. The presentations of results are well done and results are well analyzed.

As to academic writing, I suggest the author and co-authors spending more time on polishing the manuscript before getting it published.

Answer: Thank you for the general evaluation and pointing out this unclarity in the description. We have polished the manuscript by all the co-authors' prof-reading again.

Be careful with the following 3 points:

• Remember to not use long sentence if you are not sure you can handle the grammar and logic in complicated sentences.

Answer: We agree with reviewer's suggestion. We have re-written some long sentence in order to make easier to readership.

• Avoid colloquial expressions. In particular, be careful with the words such as 'maybe', 'probably', 'might' and so on. There are quite a few expressions like this throughout the manuscript. It is OK to have open questions or unresolved issues in a scientific paper, but you should state clearly what the causes/considerations are. Otherwise, it gives the readers an impressive that the results or conclusions are vague and informal and cannot be trusted.

Answer: We thank the reviewer's suggestion. We have carefully checked our statement on those points.

• Make it shorter. I would expect this paper to be concise and compact. The author should focus on the main messages that he/she wants to address to the readers. Moreover, I see it not necessary to have a separate section 'Discussion' with only the comparison of dark measurement. The discussion part is supposed to analyze and interpret the results of the study, but it is not the case here.

Answer: We agree with the reviewer. The goal of this study is to evaluate the performance of different lidar systems using the reference lidar, so we decided to compact the self-test part. The content of discussion has been re-organized into the self-test section correspondingly in the manuscript. (The similar suggestion was made out by the other reviewer).

*See Section 3.1*

Specific comments

P1L13-15: "In the end, the lidar systems were quality assured, of which the standard deviation of range-corrected signal can be controlled within 5 % at 500-2000 m while 10 % at 2000-5000 m." → One example for consideration "In the end, the lidar systems passed the quality control/assurance, ensuring that the standard deviation of range-corrected signal could be controlled within 5% at 500-2000 m and 10% at 2000-5000 m"

Answer: Thank you. We have modified in the manuscript as suggested.

The manuscript changes: "*In the end, the lidar systems passed the quality control/assurance, ensuring that the standard deviation of range-corrected signal could be controlled within 5 % at 500-2000 m and 10 % at 2000-5000 m.*"

P2L29: ' a large ground-based lidar networks': remove 'a'

Answer: It has been corrected.

P2L45: 'While to achieve', remove 'while'

Answer: It has been corrected.

P2L51-52: "Since European lidar technology was independently developed at different

stations from different countries,"-> "Since lidar systems in EARLINET were developed

independently at stations in different countries.." .

Answer: It has been corrected as suggested.

This sentence is not clear-- "the devices and algorithms used are not the same (D'Amico et al.,

2015)". EARLINET has its requirements and criteria for all the affiliated lidar stations, as well

as data processing algorithm, the Single Calculus Chain. I am not sure what devices and

algorithms you are referring to.

Answer: Thank you for pointing out this unclarify. The devices and algorithms mean to the aerosol lidars and their retrieval algorithms. The manuscript was modified accordingly to clarify this point.

The manuscript changes: "*Since lidar systems in EARLINET were developed independently at stations in different countries, the aerosol lidars and their retrieval algorithms*"

P3L80: reference missing

Answer: The missing reference has been added in the manuscript.

The manuscript changes:

*Wiegner, M. and Geiß, A.: Aerosol profiling with the Jenoptik ceilometer CHM15kx, Atmospheric Measurement Techniques, 5, 1953–1964, https://doi.org/10.5194/amt-5-1953-2012, 2012.*

P3L81: assessment of -> assess

Answer: It has been corrected.

P3L83: "At present, China is starting to build…" This sentence not clear, please revise it. Here

is an example, for your reference-- "Currently, China is in the process of building a … network,

which may lead to isolated and one-sided measurements from each observation station,

therefore…spatial-temporal correlation…"

Answer: Text was revised accordingly.

The manuscript changes:

"*Currently, China is in the process of building a comprehensive and stereoscopic observation network (Lv et al., 2020; Huang et al., 2019; Chen et al., 2019), which may lead to isolated and one-sided measurements from each observation station,*"

P3L86-87: "With the gradual shift from qualitative measurement to the qualitative application of atmospheric lidar, ... high, and direct mutual … level, China has" → As atmospheric lidars are shifting from qualitative to quantitative applications, …, increasingly high. Therefore, direct mutual…level. China has…

Answer: Text was revised accordingly.

The manuscript changes:

"*As atmospheric lidars are shifting from qualitative to quantitative applications, the requirements for its instrument function and data quality are increasingly high, and direct mutual comparison must be made at the system level.*"

P4L95: in the other region → in other regions/countries.

Answer: Text was revised accordingly.

The manuscript changes: "*in other regions or countries.*"

P4L99: This sentence is too long and lacks of clarity, please revise it. "Based on the lidar inter-comparison observation campaign on September 2021 in the south of Beijing observatory, this paper introduces the lidar quality assessment strategy based on experience of EARLINET on self-calibration and inter-comparison methods for systematic improvement of lidar hardware, and evaluates the reliability of the 1064 nm channel of many sets of lidar systems, analyzes the deviation of the Mie-Rayleigh signal and its influence on the backscatter coefficient."

Answer: The sentence has been re-written as suggested.

The manuscript changes: "*this paper introduces the lidar quality assessment strategy for 1064 nm lidar. The experience for 532 nm lidar on self-calibration and intercomparison methods by*"

*EARLINET were adopted. For systematic improvement of lidar hardware and evaluates the reliability of the 1064 nm channel of many sets of lidar systems, the deviations of the Mie-Rayleigh signal and its influence on the backscatter coefficient were analyzed. "*

P4L100: relatively, …., single wavelength -> single-wavelength

Answer: Text was revised accordingly.

The manuscript changes: " *single-wavelength*"

P4L105: on the results on the lidar signal….

Answer: Text was removed accordingly.

P4L110: , which was -> . It was

Answer: Text was revised accordingly.

P4L111: Cirrus was… and covered

Answer: Text was revised accordingly.

P4L112: in most of the time

Answer: Text was revised accordingly.

P4L113: feature -> featured

Answer: Text was revised accordingly.

P4L114: Maybe it is more accurate to say "emission and reception modules" instead of "transmitting and receiving modules ".

Answer: The text has been revised as suggested.

P4L121: The 1064 nm light -> the backscattered 1064 nm light

Answer: Text was revised accordingly.

P5L127: of which→ on which

Answer: Text was revised accordingly.

P5L129: In this sentence "The first part is self-validation or calibration according…", what are the difference between (self-) validation and calibration? If these two terms are referring to the same thing, there is no need to use different terminology here, it could confuse the readers. The authors should also check if there are confusions of terminology elsewhere in the manuscript, such as self-test, self-validation, calibration, verification… and so on.

Answer: We thank the reviewer for pointing out this inconsistency in the description.

Text was revised with a same terminology.

P5L131: CMA not defined

Answer: The manuscript was modified to explain the abbreviations: the CMA is an acronym for China meteorological administration.

P5L131-134: "With the CMA's goal of promoting the use of lidar instruments and their data among the Chinese lidar network, the inter-comparison at the hardware level was made, in terms of range corrected lidar return signals inter-compared directly, and also the inter-comparison of aerosol backscatter coefficient at 1064 nm retrieved by each lidar system was performed in this study." This sentence is not correct in grammar, please think of breaking long sentence into shorter ones for simplicity and clarity.

Answer: The sentence has been re-written.

The manuscript changes: "*The China meteorological administration (CMA) is promoting the use of lidar instruments and their data among the Chinese lidar network.To achieve this goal, the inter-comparison at the hardware level was made, in terms of the range-corrected lidar signals were inter-compared directly. And also the inter-comparison of aerosol backscatter coefficient at 1064 nm retrieved by each lidar system was performed in this study.*"

P6L137: "so that makes it …."-> This uniformity in data collection makes the inter-comparison easier.

Answer: Text was revised accordingly.

The manuscript changes: "*This uniformity in data collection makes the inter-comparison easier.*"

P6L144: of each system. Again, the sentence is too long and contains fragments.

Answer: The sentence has been re-written.

The manuscript changes: "*In order to quickly evaluate the detectable ability of each system, its signal-to-noise ratio (SNR) was assessed to estimate the detectable range using the existing method (Morille et al., 2007).*"

P6L147: due to the difference in each system efficiencies -> due to different lidar efficiencies/ Transmissions

Answer: Text was revised accordingly.

P7L150: Please specify what λr, λ0 and I are.

Answer: The manuscript was modified accordingly to clarify all these symbols.

P7L151: please revise this sentence "It should be compared with simultaneous observation,

and continuously collecting the original data for at least 180 minutes, and selecting a period of no

less than 30 minutes where the aerosol vertical distributions are stable."

Answer: Text was revised accordingly.

The manuscript changes: "*Which $\lambda_r$ is the wavelength of lidar signal; i is the number of lidar systmes. Because the comparison should be made between simultaneous observations, so a period of no less than 30 minutes was selected during a continuous observation for at least minutes, and also the aerosol vertical distributions are relatively stable during the selecting period. The rang corrected signal is obtained by using the raw collection data after the cumulative average, background subtraction, and rang correction.*"

P7L152: "The rang-square-correction signal" what is it? Range-corrected signal is commonly

used in the lidar community.

Answer: Text was revised accordingly.    "Range-corrected signal" has been used consistently in the whole manuscript.

P7L157: No. Ref lidar is confusing, better call it reference lidar for clarity.

Answer: Text was revised as suggested.

P7L170: I do not see the link between signal saturation and fitting molecular signal to RCS.

please explain.

Answer: The sentence has been re-written to clarify this point.

Because the effects caused by saturation of detectors can be found from lidar signal intensity, thus we decided to fit the molecular attenuated backscatter coefficient to the RCS of each lidar system in this study.

The manuscript changes: "*Because the effects caused by saturation of detectors can be found from lidar signal intensity, thus we decided to fit the molecular attenuated backscatter coefficient to the RCS of each lidar system in this study.*"

P7L172: "…a good signal-to-noise ratio except No. L05 lidar adopted…". Break into 2 sentence

"…a good signal-to-noise ratio. However, No. L05 lidar adopted….at such range"

Answer: Text was revised accordingly.

The manuscript changes: *"However, No. L05 lidar adopted the normalization range between 12000m to 13000m due to its signal distortion at the range from 3000m to 7000 m."*

P7L174: typo: a space missing 'than' and '5%'

Answer: Text was revised accordingly.

P8L177--L179: Could you give more comment about the data quality of lidar No.05? Is it possible to correct this electronic noise? Could this mismatch with Rayleigh fit be caused by other reasons, for example, the divergence of laser beam? Have you ever got better Rayleigh fit from this lidar?

Answer: The analog signal from APD commonly suffers from the electronic noise.

Yes, it is possible to correct. A method to correct such issue was proposed by Freudenthaler, V. et.al. 2018, but it has to be done after each observation. It might be the other reason caused by the divergence of laser beam, however we are not able to check. Due to the very limited measurement period, we have not got better Rayleigh fit from this lidar.

 Freudenthaler, V., Linné, H., Chaikovski, A., Rabus, D., and Groß, S.: EARLINET lidar quality assurance tools, Atmospheric Measurement

Techniques Discussions, pp. 1–35, 2018.

P8L180: Better to write "In this test, lidar profiles in every 30-minute time intervals were averaged."

Answer: Text was revised accordingly.

P9 Figure 4: To me, raw lidar signal means it is without any correction, such as deadtime, bin shift, etc. In Figure 9, I guess corrections have been applied.

Answer: Yes, in figure 9, the corrections haven been applied. We agree with the reviewer, so the "raw lidar signal" changes to "corrected lidar signal by deadtime, bin shift corrections"

The manuscript was modified to clarify these points.

P10L193: This is a bit confusing, please rephrase it "which also means they can be used to observe the vertical distributions of aerosol and cloud without knowing the determining the amounts".

Answer: The sentence has been re-written for clarify.

The manuscript changes: *"Therefore The results indicated that the signal from each lidar system are comparable qualitatively, and also they are able to observe the vertical distributions of aerosol*

*and cloud relatively during the temporal and space evolution.*"

P10L194: quantitative analysis→ quantitatively analyze

Answer: Text was revised accordingly.

P10L194: it is not proper to call it 'accuracy', better to use 'difference'

Answer: We agree with the reviewer. Text was revised accordingly.

P10L195: overlap properties may cause misunderstanding, you can simply say 'overlap ranges'

Answer: Text was revised as suggested.

P10L197: "In this investigation, single…..was selected"→ "Lidar observations between 18:30 and 19:00 CST from each lidar system were averaged for inter-comparison. "

Answer: Text was revised as suggested.

The manuscript changes: "*Lidar observations between 18:30 and 19:00 CST from each lidar system were averaged for inter-comparison.*"

P10L198: From the quicklooks, the readers can see the presence of aerosol layers in the range of 1500-2000, why did you normalize the lidar signal in the aerosol layer although you have mentioned in the previous sentence that the range 2000-5000 was expected to be clean?

What if you normalize at higher altitude, should the results be better?

Answer: We thank the reviewer for pointing out this miss-understanding.

The reason to select the range of 1500-2000 is because the signal distortion of No.L05 lidar at the range between 2000 m and 5000 m.

The manuscript was modified accordingly to clarify this point. "*and also it has less effect by the lidar system at such range.*"

P11L202-L203: A few grammar mistakes in this sentence. Please rewrite it, here is one example "Due to the significant differences in the incomplete overlap region between different lidar systems, large relative deviations were observed within the 500 m range (Figure 6, b). As a result, a meaningful comparison cannot be made"

Answer: Text was revised as suggested.

About P10 Figure 5:

■ The ticks of Figure 5(c,d) are partly cut off, please replace with complete figures

Answer: Plot was replaced as suggested.

■ The structures within boundary layer looks similar in all quicklooks except for Figure

5(e), could you comment on that?

Answer: The structures within boundary layer were same, but only the color bar were scaled

differently.

■ Where do the repeating white stripes in Figure 5(b) come from? Is it a technical

anomaly?

Answer: Yes, it is a technical anomaly due to some operation issues.

The manuscript was modified accordingly to clarify this point.

■ In Figure 5(a) and 5(c), we see ripples in the transported aerosol layer below 3000 m

from 19:00 to 23:00, can you comment on that?

Answer: the ripples were caused by unstable laser energy due to the rapid temperature change.

P11, P12: Figure 6(c) and 7(c) do not carry so much information, therefore not necessary.

Answer: Figure 6(c) and 7(c) present the mean standard deviation in the high aerosol loading and

low aerosol loading distributed range. Such information is useful to avoid the effect by the spatial

variance. The manuscript was modified accordingly to clarify this point.

The manuscript changes: ""*In order to avoid the effect by the spatial variance, the mean relative
deviations were also presented (Figure 7,c)*."

P13: Please write how you determine system noise and random noise and put references

The reviewer stops listing corrections of English writing from Page 11 and advices the author

and coauthors to correct it by themselves.

Answer: The random noise is determined by mean standard deviation of background noise, and the

system noise is determined by mean standard deviation of random noise of all the ranges.

The manuscript was modified accordingly to clarify this point:"*The random noise is determined
by the mean standard deviation of background noise, and the system noise is determined by the
mean standard deviation of random noise of all the ranges.*"

Figure 8: (1) The scales of Y-axis in Figure 8(a) and Figure 8(b) are so different, making it

difficult to conclude. (b) For Figure 8(b), did you take multiple dark measurements for L05 in

order to check if such noises are stable or not?

Answer: The dimensions between analog and PC mode are not comparable, so the scales are

meaningless, but the structure variation with the range is an indicator to judge the data

quality.   The systemic noise was always exist in this lidar.

P14L267: "we found … also such lidar system has a possibility to be miniaturization". It is not appropriate to draw such conclusion, because this paper does not talk about the miniaturization.

Answer: We agree with the reviewer. So we decided to remove this sentence.

To reviewer 2

**Comments:**

The Wang et al manuscript entitled "Quality assessment of aerosol lidars at 1064 nm in the framework of the MEMO campaign" presents the quality assessment of lidar performances at 1064 nm in order to gain the confidence for establishing lidar network in China. The experiment and results in this study could be a useful experience for regularly lidar quality assessment in a large lidar network, in particular the 1064 nm lidar have been widely used however there is very few reports regarding on their hardware assessment a lidar network. Therefore, to my opinion, it can be published in AMT after minor revision.   I have some comments as following.

Answer: Thank you for the general evaluation and pointing out this unclarity in the description.

The specific comments are listed below:

I recommend the authors to provide the differences in the calibration procedures between 532 nm and 1064 nm channels.

Answer: Thank you for the suggestion.

In principle both calibration procedures should be same, however, due to the weaker molecular signal-to-noise ratio (SNR) at 1064 nm compared to 532 nm for these instruments, it is very challenging to find the pure molecular signal as a reference at 1064 nm, so the calibration of 1064 nm channels were made by using the calibrated 532 nm signal in the previous.

In this study, due to the better efficiency of SPAD detection, the high SNR can be obtained in 1064 nm channels, so the adopted calibration procedures for 1064 nm channels are the same with the calibration procedures for 532 nm channels by EARLINET in the previous.

This point has been provided in the manuscript:"*it is very challenging to find the pure molecular signal as a reference at 1064 nm, so calibration for 1064 nm attenuated total backscatter (ATB) are based on the 532 nm ATB calibration (Vaughan et al., 2019)."*

If possible, the dark noise test, telecover test, detection range test and Rayleigh fitting test of the six lidars should also be summarized and compared in the paragraph.

Answer: We agree with the reviewer for the suggestion, however the telecover test experiments

were done by each individual manufactures, we are not able to collect these complete dataset.

Moreover, the goal of this study is to evaluate the performance of different lidar systems using the reference lidar, which is based on the self-checks were assumed to be done. Therefore, we decide to summarize the dark noise test, detection range test and Rayleigh fitting test of the reference lidar as an example. In addition, the dark noise test were compared between the analog mode and photon counting mode.

The manuscript was modified accordingly: see section 3.1.

The Discussion section should be omitted and the dark measurements results should be analyzed in the self-test section.

Answer: We agree with the reviewer. The related content has been re-organized into the self-test section correspondingly in the manuscript. (The similar suggestion was made out by reviewer 1, see section 3.1.).

The technical corrections:

P3L70: "at 532 nm " to "at 355 nm and 532 nm "

Answer: this point has been corrected in the manuscript.

P3L74: "instruments, calibration for 1064 nm attenuated total backscatter (ATB) calibration are..." remove one "calibration "

Answer: This sentence has been re-written in the manuscript.

*"calibration for 1064 nm attenuated total backscatter (ATB) are based on the 532 nm*

*ATB calibration*"

P3L80: reference missing?

**Answer:** We thank the reviewer for pointing out this careless. The missing reference is added in the manuscript.

Wiegner, M. and Geiß, A.: Aerosol profiling with the Jenoptik ceilometer CHM15kx, Atmospheric Measurement Techniques, 5, 1953–1964, https://doi.org/10.5194/amt-5-1953-2012, 2012.

P4L111: "Cirrus is" to "Cirrus was"

**Answer:** It has been corrected in the manuscript.

P4L113: "infrared channels" to "infrared Mie-Rayleigh channels"

**Answer:** It has been corrected in the manuscript.

P4L115: please check the grammer with the sentence

**Answer:** The grammar has been corrected in the manuscript.

"*The ID numbers were made up for each lidar system at 1064nm for easier identification and their hardware parameters were provided by their manufacturers, which are summarized in Table 1.*"

P5L130: the results of "telecover test" is missing

**Answer:** We modified the manuscript to clarify this (see the answer for the general comment).

P5L131: please define the abbreviation of "CMA"

**Answer:** We thank the reviewer for pointing out this inconsistency. The manuscript was modified to explain the abbreviations: the CMA is an acronym for China meteorological administration.

P6L138: since the authors uploaded the automatic "Atmospheric Lidar Evaluation program (ALiE, https://gitee.com/mualidar/cma-lidar-comparison)" somewhere, I suggest also document this work into AMT's supplement if possible, so that it could be more benefit for the lidar community.

**Answer:** We agree the reviewer's suggestion. We could upload the code as supplement.

P6L147: what does "system efficiencies" indicate?

**Answer:** We mean lidar efficiencies/transmissions.

P7L153-154: re-write this sentence

**Answer:** We have re-written this sentence in order to make better sense:"*Which $\lambda r$ is the wavelength of lidar signal; i is the number of lidar systmes. Because the comparison should be made between simultaneous observations, so a period of no less than 30 minutes was selected during a continuous observation for at least 180 minutes, and also the aerosol vertical distributions are relatively stable during the selecting period. The rang corrected signal is obtained by using the raw collection data after the cumulative average, background subtraction, and rang correction.*"

P7L160: should symbol S be indicates not only "lidar signal" but also backscatter coefficient?

**Answer:** We thank the reviewer for pointing out this unclarify. Yes, it should be both lidar signal and aerosol backscatter coefficient. It has been corrected in the manuscript.

Equation 3: "δ" to "δi"

**Answer:** It has been corrected in the manuscript.

P7L173: it is unclear what does "signal issue" mean?

**Answer:** It means the lidar signal at the range of 6000-7000 has distortion.

It has been explain in the manuscript.

P8L178: "can be" to "can not be"

**Answer:** It has been checked and corrected in the manuscript.

P8L189: define "CST"

**Answer:** We thank the reviewer for pointing out this inconsistency. The manuscript was modified to explain the abbreviations: the CST is an acronym for Central Standard Time.

P10L194: unclear description

**Answer:** It has been re-writen in the manuscript.

P11L203: "it is" to "they are"

**Answer:** It has been corrected in the manuscript.

P12L224: "got severe distortion" means?

**Answer:** It has been corrected in the manuscript.

P13L249: "2017" to "2021"?

**Answer:** The typo has been corrected in the manuscript.

Reference L347-348:

Please update the preprint to the published one:

Mamouri, R.-E. and Ansmann, A.: Potential of polarization/Raman lidar to separate fine dust, coarse dust, maritime, and anthropogenic aerosol profiles, Atmos. Meas. Tech., 10, 3403–3427, https://doi.org/10.5194/amt-10-3403-2017, 2017.

**Answer:** The reference has been replaced in the manuscript.